# GLOBAL IDENTIFIABILITY OF OVERCOMPLETE DICTIONARY LEARNING VIA L1 AND VOLUME MINIMIZATION

**Yuchen Sun & Kejun Huang**
Department of Computer and Information Science and Engineering
University of Florida
Gainesville, FL 32611, USA
`{yuchen.sun,kejun.huang}@ufl.edu`

## ABSTRACT

We propose a novel formulation for dictionary learning with an overcomplete dictionary, i.e., when the number of atoms is larger than the dimension of the dictionary. The proposed formulation consists of a weighted sum of $\ell_1$ norms of the rows of the sparse coefficient matrix plus the log of the matrix volume of the dictionary matrix. The main contribution of this work is to show that this novel formulation guarantees global identifiability of the overcomplete dictionary, under a mild condition that the sparse coefficient matrix satisfies a strong scattering condition in the hypercube. Furthermore, if every column of the coefficient matrix is sparse and the dictionary guarantees $\ell_1$ recovery, then the coefficient matrix is identifiable as well. This is a major breakthrough for not only dictionary learning but also general matrix factorization models as identifiability is guaranteed even when the latent dimension is higher than the ambient dimension. We also provide a probabilistic analysis and show that if the sparse coefficient matrix is generated from the widely adopted sparse-Gaussian model, then the $m \times k$ overcomplete dictionary is globally identifiable if the sample size is bigger than a constant times $(k^2/m) \log(k^2/m)$ with overwhelming probability. Finally, we propose an algorithm based on alternating minimization to solve the new proposed formulation.

## 1 INTRODUCTION

Dictionary learning (DL) amounts to factor a data matrix as $X = AS$ where $S$ is sparse (Tošić & Frossard, 2011), which may also be known as sparse coding (Olshausen & Field, 1997) or sparse component analysis (Georgiev et al., 2005) in various fields. Treating $X \in \mathbb{R}^{m \times n}$ or $\mathbb{C}^{m \times n}$ as a collection of data samples as its columns, this factorization means that each sample is a *sparse* combination of the columns of $A$, or in other words atoms of the dictionary. Unlike the task of compressive sensing or sparse vector recovery, in which case the dictionary matrix $A$ is given, dictionary learning tries to find both $A$ and $S$, therefore the problem is a lot more challenging. Depending on the shape of the dictionary matrix $A$, we may seek to find a complete dictionary if $A$ is square or an overcomplete dictionary if $A$ is wide. In this paper we focus on overcomplete dictionary learning, therefore $A \in \mathbb{R}^{m \times k}$ and $S \in \mathbb{R}^{k \times n}$ (or $\mathbb{C}^{m \times k}$ and $\mathbb{C}^{k \times n}$, respectively) where $m < k$.

Dictionary learning has found numerous applications in signal denoising (Elad & Aharon, 2006), audio coding (Plumbley et al., 2009), and medical imaging (Tošić et al., 2010), to name just a few. On the theory side, most of the existing works have focused on algorithm design. Famous algorithms include $k$-SVD (Aharon et al., 2006a) and online dictionary learning (Mairal et al., 2009), among numerous other algorithms based on generic nonconvex algorithm design with guarantee of convergence to a stationary point. More recently, there has appeared a line of research that attempts to show global optimality for dictionary learning under more restrictive assumptions, such as (Spielman et al., 2012; Agarwal et al., 2016; Arora et al., 2014; 2015; Sun et al., 2016a;b; Rambhatla et al., 2019; Bai et al., 2019; Zhai et al., 2020a;b; Shen et al., 2020; Tolooshams & Ba, 2022).

## 1.1 Prior work on identifiability of DL

A matrix factorization model without any additional assumptions on the latent factors is known to be not unique, since we can always "insert" an invertible matrix $W$ and $W^{-1}$ as $X = \widetilde{A}\widetilde{S}$ where $\widetilde{A} = AW^{-1}$ and $\widetilde{S} = WS$, and one cannot distinguish whether $S$ or $\widetilde{S}$ are the groundtruth sources. Furthermore, if the dictionary $A$ is overcomplete with $m < k$, columns of $\widetilde{S}$ could include additional components that are orthogonal to the row space of $A$, i.e., $\widetilde{S} = W(S + B)$ where $AB = 0$, while we still have $X = AS = AW^{-1}W(S + B) = \widetilde{A}\widetilde{S}$. If, however, a learning criterion $q(A, S)$ is imposed so that the resulting ambiguities can only be permutations and scaling, then we say the model is identifiable, as is formalized as follows. Notice that we differentiate the identifiability of just the dictionary $A$ and the whole factorization model, which is indeed not equivalent when the dictionary is overcomplete.

**Definition 1** (Identifiability). Consider the generative model $X = A_\natural S_\natural$, where $A_\natural$ and $S_\natural$ are the groundtruth latent factors. Let $(A_\star, S_\star)$ be optimal for an identification criterion $q$

$$(A_\star, S_\star) = \underset{X=AS}{\arg\min}\, q(A, S).$$

If $A_\natural$ and/or $S_\natural$ satisfy some condition such that for any $(A_\star, S_\star)$, there exist a permutation matrix $\Pi$ and a diagonal matrix $D$ such that $A_\natural = A_\star D\Pi$, then we say $A_\natural$ is essentially identifiable, up to permutation and scaling, under that condition; if we further have that $S_\natural = \Pi^\top D^{-1} S_\star$, then we say that the matrix factorization model is essentially identifiable, up to permutation and scaling, under that condition.

When dictionary learning was first proposed, a common learning criterion $q(A, S)$ is simply the total number of nonzeros in $S$, sometimes also called the $\ell_0$ (pseudo-)norm $\|S\|_0$. If every column of $S_\natural$ is $s$-sparse, then via some combinatorial calculation, it has been shown that the $\ell_0$-norm minimization criterion guarantees identifiability if the spark of $A$ is at least $2s$ and the sample size $n$ is $O((s+1)\binom{k}{s})$ (Aharon et al., 2006b; Hillar & Sommer, 2015; Garfinkle & Hillar, 2019). The main drawback is that the required sample size $n$ is usually too large to be practical. Cohen & Gillis (2019) reduced the sample complexity down to $O(k^3/(k-s)^2)$, but also restricted the dictionary to be complete, i.e., $m \geq k$.

Another famous learning criterion for DL, inspired by the success of compressive sensing (Donoho, 2006; Candès & Wakin, 2008), is the following formulation with $q(A, S)$ being the summation of the absolute values of $S$ plus indicator functions that columns of $A$ have bounded $\ell_2$ norms:

$$\underset{A,S}{\text{minimize}}\quad \|S\|_1 \qquad \text{subject to}\quad X = AS, \|Ae_c\|_2 \leq 1, c = 1, \ldots, k. \qquad (1)$$

Identifiability results based on the $\ell_1$ norm formulation have been predominantly local, meaning the model is identifiable within a neighborhood of the groundtruth factors $(A^\natural, S^\natural)$, while the dictionary is restricted to be complete (and incoherent) (Gribonval & Schnass, 2010; Wu & Yu, 2017; Wang et al., 2020), with the sole exception of (Geng & Wright, 2014) for overcomplete DL. The advantage is that the sample size requirement is typically down to $O(k \log k)$ and allows the existence of dense outliers. Global identifiability is achieved by Hu & Huang (2023a); Sun & Huang (2024) by using a matrix volume criterion $|\det A|$ while constraining the $\ell_1$ norms of the rows of $S$ with the same sample complexity, although as the criterion suggests it only applies to complete dictionaries.

## 1.2 This paper

In this paper, we propose the following novel formulation for overcomplete dictionary learning, and show that global identifiability can be achieved under mild conditions:

$$\underset{A,S}{\text{minimize}}\quad \frac{1}{2}\log\det AA^\top + \underset{\|d\|_2^2=m}{\max}\sum_{c=1}^{k} d_c\|e_c^\top S\|_1 \qquad \text{subject to}\quad X = AS \qquad (2)$$

This means the learning criterion $q(A, S)$ consists of two parts: a weighted sum of the $\ell_1$ norms of the rows of $S$ and a term that is proportional to the "volume" of the dictionary matrix $A$—for an overcomplete dictionary, the volume is defined as $\det AA^\top$ (Ben-Israel, 1992). Our contributions are as follows:

1. We give a deterministic characterization of global identifiability of overcomplete dictionary learning via solving (2). Our analysis shows that a sufficient condition is that 1) every column of $S_\natural$ is at most $s$-sparse and $A$ is a dictionary that guarantees exact recovery of all $s$-sparse vectors via $\ell_1$ minimization, and 2) the cellular hull of $S_\natural$ is $m$-sufficiently scattered in the $k$-hypercube $[-1, 1]^k$. The resulting identifiability condition is almost minimal: the first condition is obviously necessary as otherwise $S_\natural$ would not be identifiable even if the overcomplete $A_\natural$ is correctly recovered; the second condition is a slightly stronger condition than that of complete dictionary learning. It is appealing to see that no other conditions are needed to guarantee global identifiability.

2. We further provide a probabilistic characterization of when a randomly generated factor matrix satisfies the aforementioned identifiability conditions. Since we only require $A_\natural$ to guarantee exact recovery of $s$-sparse vectors via $\ell_1$ minimization, for which there exist numerous work on this topic (such as when an i.i.d. Gaussian matrix satisfies the restricted isometry property with high probability), we will be focusing on studying the sample complexity of $S$. We adopt the sparse-Gaussian model, i.e., every column contains at most $s$ nonzero values that are drawn from i.i.d. standard normal and show that the resulting $S$ satisfies the $m$-strongly scattered in the $k$-hypercube $[-1, 1]^k$ with overwhelming probability if $k < m$ and $n$ is $O((k^2/m) \log(k^2/m))$. Notice that it is again a sharp generalization of complete dictionary learning with sample complexity $O(k \log(k))$ (Hu & Huang, 2023a), and a factor of $k$ better than the works that focus on global optimality guarantees of overcomplete DL (Agarwal et al., 2016; Rambhatla et al., 2019).

3. We propose an alternating minimization algorithm for the novel identification criterion of overcomplete DL. The formulation is first modified slightly by moving the exact factorization constraint as a data fidelity term in the objective function, then the overcomplete dictionary and the sparse coefficient matrices are updated alternatingly via a gradient-type step. As the problem is NP-hard, no known algorithm is able to guarantee convergence to a global optimum. The proposed algorithm is applied to synthetically generated data to demonstrate that global identifiability can indeed be guaranteed via solving (2).

## 2 IDENTIFIABILITY ANALYSIS

In this section, we provide analysis on when solving (2) guarantees the exact recovery of the overcomplete dictionary $A_\natural$ and/or the sparse coefficient matrix $S_\natural$. In Definition 1 we mentioned that it is acceptable to recover $A_\natural$ up to column permutation and scaling. While column permutation does not affect the objective value of (2), column scaling does. Therefore, we first study the optimal scaling that is induced from solving (2), which provides important insights into the subsequent analysis of identifiability. We provide both a deterministic condition and a probabilistic generative model that guarantees identifiability with overwhelming probability.

### 2.1 OPTIMAL SCALING

**Lemma 1.** *Let* $(A_\star, S_\star)$ *be an optimal solution of* (2)*, then*

$$\|e_c^\top S_\star\|_1 = \sqrt{\left[A_\star^\top \left(A_\star A_\star^\top\right)^{-1} A_\star\right]_{cc}} = d_{\star c}, \qquad c = 1, \dots, k. \tag{3}$$

*where* $d_{\star c}$ *are the optimal weights that reach the maximum of* $\sum_c d_c \|e_c^\top S_\star\|_1$.

*Proof.* If $(A_\star, S_\star)$ is feasible for (2), then so is $(A_\star \Psi, \Psi^{-1} S_\star)$ where $\Psi$ is a diagonal matrix with $c$th diagonal entry $\psi_c$. Plugging $(A_\star \Psi, \Psi^{-1} S_\star)$ into the objective of (2) and optimize with respect to $\Psi$ while fixing $(A_\star, S_\star)$, then $\Psi = I$ should be an optimal solution. Taking the derivative with respect to $\psi_c$ and setting it equal to zero, we get

$$a_c^\top \left(A_\star \Psi^2 A_\star^\top\right)^{-1} a_c \psi_c - d_c \|e_c^\top S_\star\|_1 / \psi_c^2 = 0,$$

where $a_c^\top$ is the $c$th row of $A_\star$. If $\psi_c = 1$ is optimal, then we must have

$$d_c \|e_c^\top S_\star\|_1 = \left[A_\star^\top \left(A_\star A_\star^\top\right)^{-1} A_\star\right]_{cc}. \tag{4}$$

To maximize $\sum_c d_c \|\boldsymbol{e}_c^\top \boldsymbol{S}_\star\|_1$ subject to $\|\boldsymbol{d}\|_2^2 = m$, we know from the Cauchy-Schwarz inequality that $d_{\star c}$ should be chosen as some scalar $\alpha$ times $\|\boldsymbol{e}_c^\top \boldsymbol{S}_\star\|_1$ such that

$$\|\boldsymbol{d}_\star\|_2^2 = \sum_{c=1}^k \alpha^2 \|\boldsymbol{e}_c^\top \boldsymbol{S}_\star\|_1^2 = m.$$

Plugging $d_{\star c} = \alpha \|\boldsymbol{e}_c^\top \boldsymbol{S}_\star\|_1$ into (4) and sum over $c = 1, \ldots, k$ shows

$$\sum_{c=1}^k \alpha \|\boldsymbol{e}_c^\top \boldsymbol{S}_\star\|_1^2 = \sum_{c=1}^k \left[ \boldsymbol{A}_\star^\top \left( \boldsymbol{A}_\star \boldsymbol{A}_\star^\top \right)^{-1} \boldsymbol{A}_\star \right]_{cc} = \operatorname{Tr} \boldsymbol{A}_\star^\top \left( \boldsymbol{A}_\star \boldsymbol{A}_\star^\top \right)^{-1} \boldsymbol{A}_\star = \operatorname{Tr} \left( \boldsymbol{A}_\star \boldsymbol{A}_\star^\top \right)^{-1} \boldsymbol{A}_\star \boldsymbol{A}_\star^\top = m.$$

This means $\alpha = 1$, and therefore (3) holds. $\qquad\square$

## 2.2 IDENTIFIABILITY ANALYSIS

**Assumption 1.** The columns of $\boldsymbol{A}_\natural$ and rows of $\boldsymbol{S}_\natural$ are scaled and counter-scaled to satisfy:

$$\|\boldsymbol{e}_c^\top \boldsymbol{S}_\natural\|_1 = \sqrt{\left[ \boldsymbol{A}_\natural^\top \left( \boldsymbol{A}_\natural \boldsymbol{A}_\natural^\top \right)^{-1} \boldsymbol{A}_\natural \right]_{cc}}, \qquad c = 1, \ldots, k. \tag{5}$$

**Assumption 2.** Rows of $\boldsymbol{A}_\natural$ and $\boldsymbol{S}_\natural$ are both linearly independent. Matrix $\boldsymbol{A}_\natural$ does not contain zero columns.

**Assumption 3** ($m$-strongly scattered in the $k$-hypercube). Let $C_k$ denote the $k$-hypercube $C_k = \{\boldsymbol{x} \in \mathbb{R}^k \mid \|\boldsymbol{x}\|_\infty \le 1\}$. Define $\mathcal{B}_m$ as the following set

$$\mathcal{B}_m = \left\{ \operatorname{Diag}(\|\boldsymbol{q}_1\|_2, \ldots, \|\boldsymbol{q}_k\|_2)^\dagger \boldsymbol{Q} \boldsymbol{p} \mid \forall\, \boldsymbol{Q} \in \mathbb{R}^{k \times m} : \boldsymbol{Q}^\top \boldsymbol{Q} = \boldsymbol{I}, \, \boldsymbol{p} \in \mathbb{R}^m : \|\boldsymbol{p}\|_2 = 1 \right\},$$

where $\boldsymbol{q}_c$ denotes the $c$th row of $\boldsymbol{Q}$. A set $\mathcal{S} \in \mathbb{R}^k$ is $m$-strongly scattered in the $k$-hypercube if:

1. $\mathcal{B}_m \subseteq \mathcal{S} \subseteq C_k$;

2. $\partial \mathcal{B}_m \cap \partial \mathcal{S} = \{\operatorname{Diag}(\|\boldsymbol{q}_1\|_2, \ldots, \|\boldsymbol{q}_k\|_2)^\dagger \boldsymbol{Q} \boldsymbol{q} / \|\boldsymbol{q}\|_2 \mid \boldsymbol{q} \text{ are rows of } \boldsymbol{Q} \text{ with } \boldsymbol{Q}^\top \boldsymbol{Q} = \boldsymbol{I}\}$, where $\partial$ denotes the boundary of the set.

The definition of the set $\mathcal{B}_m$ involves all $k \times m$ matrices with orthonormal columns $\boldsymbol{Q}$ and normalized $m$-vectors $\boldsymbol{p}$. To see that $\mathcal{B}_m$ is indeed a subset of $C$, notice that the rows of $\boldsymbol{\Psi} \boldsymbol{Q}$ are either zero or unit norms by construction, so all elements of the vector $\boldsymbol{\Psi} \boldsymbol{Q} \boldsymbol{p}$ are in $[-1, 1]$ using Cauchy-Schwarz inequality. Assumption 3 is equivalent to the sufficiently scattered condition proposed by Hu & Huang (2023a) when $m = k$, and more restrictive when $m < k$. As we will see, such restriction will be useful to establish identifiability for overcomplete DL. The sufficiently scattered condition has many variations for various identifiable unsupervised learning models, such as nonnegative matrix factorization (Huang et al., 2013; Fu et al., 2015; Huang et al., 2016; 2018), simplicial representation learning (Fu et al., 2015; Lin et al., 2015; Huang & Fu, 2019), and bounded component analysis (Tatli & Erdogan, 2021; Hu & Huang, 2023b; 2024). However, to the best of our knowledge, this is the first variation that is capable of guaranteeing identifiability when the latent dimension is *higher* than the ambient dimension. An illustration of 2-strongly scattered in the 3-hypercube is shown on the right of Figure 1, compared with 3-strongly scattered on the left, which is equivalent to the sufficiently scattered condition presented by Hu & Huang (2023a).

Assumption 3 will be imposed on the cellular hull of $\boldsymbol{S}_\natural$, which is defined as follows:

**Definition 2** (Cellular hull). The cellular hull of a finite set of vectors $\{\boldsymbol{s}_1, \ldots, \boldsymbol{s}_n\}$, stacked as the columns of the matrix $\boldsymbol{S}$, is

$$\operatorname{cell}(\boldsymbol{S}) = \left\{ \boldsymbol{S} \boldsymbol{\theta} \,\middle|\, \|\boldsymbol{\theta}\|_\infty \le 1 \right\}.$$

Consider the groundtruth sparse coefficient matrix $\boldsymbol{S}_\natural$, if we rescale its rows to have unit $\ell_1$ norms, denoted as $\widetilde{\boldsymbol{S}}_\natural$, then $\operatorname{cell}(\widetilde{\boldsymbol{S}}_\natural) \subseteq C_k$ due to Hölder's inequality $|\boldsymbol{a}^\top \boldsymbol{b}| \le \|\boldsymbol{a}\|_1 \|\boldsymbol{b}\|_\infty$. For identifiability of overcomplete DL, we would require $\operatorname{cell}(\widetilde{\boldsymbol{S}}_\natural)$ to be $m$-strongly scattered in the $k$-hypercube, as formally stated as follows:

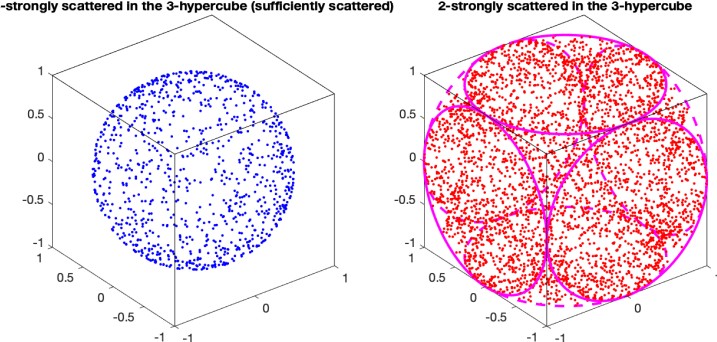

Figure 1: An illustration of $\mathcal{B}_2$ in $\mathbb{R}^3$ on the right. In comparison, $\mathcal{B}_3$ in $\mathbb{R}^3$ is the Euclidean ball illustrated on the left. While $\mathcal{B}_3$ touches the boundary of $[-1, 1]^3$ at only 6 points $\pm e_1, \pm e_2$, and $\pm e_3$, $\mathcal{B}_2$ touches each face of $[-1, 1]^3$ at a circle with radius 1, as shown in magenta on the right. In the context of dictionary learning, a $3 \times 3$ complete dictionary is identifiable if $\mathrm{cell}(S_\natural) \supseteq \mathcal{B}_3$, shown on the left, while a $2 \times 3$ overcomplete dictionary is identifiable if $\mathrm{cell}(S_\natural) \supseteq \mathcal{B}_2$, shown on the right.

**Theorem 1.** *Consider the overcomplete DL model $X = A_\natural S_\natural$, where $A_\natural \in \mathbb{R}^{m \times k}$ is the groundtruth mixing matrix and $S_\natural \in \mathbb{R}^{k \times n}$ is the groundtruth sparse coefficient matrix. Suppose $A_\natural$ and $S_\natural$ satisfies Assumptions 1 and 2. Furthermore, let $\widetilde{S}_\natural$ denote the matrix obtained from rescaling the rows of $S_\natural$ to have unit $\ell_1$ norms, and assume that $\mathrm{cell}(\widetilde{S}_\natural)$ is m-strongly scattered in the k-hypercube. Then for any solution of (2), denoted as $(A_\star, S_\star)$, there exist a permutation matrix $\Pi$ and a diagonal matrix $D$ such that $A_\natural = A_\star D\Pi$. In other words, an overcomplete dictionary $A_\natural$ is identifiable if the groundtruth $A_\natural$ and $S_\natural$ satisfies Assumptions 1, 2, and $\mathrm{cell}(\widetilde{S}_\natural)$ satisfies Assumption 3.*

*Proof sketch.* Assumption 2 asserts that rows of $A_\natural$ are linearly independent, so there exists a $k \times m$ matrix $Q$ with orthonormal columns that spans the row space of $A_\natural$, then $Q^\top Q = I$ and

$$QQ^\top = A_\natural^\top \left(A_\natural A_\natural^\top\right)^{-1} A_\natural.$$

Using the diagonal matrix $D_\natural$, which is defined as

$$[D_\natural]_{cc} = \sqrt{\left[A_\natural^\top \left(A_\natural A_\natural^\top\right)^{-1} A_\natural\right]_{cc}}, \qquad c = 1, \ldots, k, \tag{6}$$

and $q_c^\top$ as rows of $Q$ defined in Assumption 3, we have

$$[D_\natural]_{cc} = \|e_c^\top S_\natural\|_1 = \sqrt{[QQ^\top]_{cc}} = \|q_c\|_2. \tag{7}$$

For two wide $k \times n$ matrices $S_\natural$ and $S_\star$, there exist a $k \times k$ matrix $W$ and $k \times n$ matrix $B$ such that

$$S_\star = WS_\natural + B,$$

where $W = S_\natural^\dagger S_\star$ and rows of $B$ are in the null space of $S_\natural$, i.e., $S_\natural B^\top = 0$. Denote $w_c^\top$ and $b_c^\top$ as the $c$th row of $W$ and $B$, respectively, then

$$\|e_c^\top S_\star\|_1 = \|w_c^\top S_\natural + b_c^\top\|_1 \geq w_c^\top S_\natural \theta + b_c^\top \theta = w_c^\top D_\natural \widetilde{S}_\natural \theta + b_c^\top \theta, \quad \forall \|\theta\|_\infty \leq 1.$$

If $\mathrm{cell}(\widetilde{S}_\natural)$ is $m$-strongly scattered in the $k$-hypercube, then for all $\|p\|_2 = 1$ we can find a $\|\theta\|_\infty$ such that $D_\natural^{-1} Qp = \widetilde{S}_\natural \theta$. In Appendix A we show that $\|e_c^\top S_\star\|_1 = \|w_c^\top S_\natural\|_1$, as a result,

$$\|e_c^\top S_\star\|_1 = \|w_c^\top S_\natural\|_1 = \|w_c^\top D_\natural \widetilde{S}_\natural\|_1 \geq w_c^\top D_\natural D_\natural^{-1} Qp = w_c^\top Qp, \quad \forall \|p\|_2 = 1.$$

Choose $p = Q^\top w_c / \|Q^\top w_c\|_2$, we have $\|e_c^\top S_\star\|_1 \geq \|Q^\top w_c\|_2$. Square both sides and sum over $c = 1, \ldots, k$, we get

$$\|Q^\top W^\top\|_F^2 = \sum_{c=1}^{k} \|Q^\top w_c\|_2^2 \leq \sum_{c=1}^{k} \|e_c^\top S_\star\|_1^2 = \mathrm{Tr}\, A_\star^\top \left(A_\star A_\star^\top\right)^{-1} A_\star = m. \tag{8}$$

Since both $(A_\natural, S_\natural)$ and $(A_\star, S_\star)$ are feasible, $X = A_\natural S_\natural = A_\star S_\star = A_\star(WS_\natural + B)$. Multiplying both sides by $S_\natural^\dagger$ gives

$$A_\natural = A_\natural S_\natural S_\natural^\dagger = A_\star(WS_\natural + B)S_\natural^\dagger = A_\star W.$$

Then we have

$$\log \det A_\star A_\star^\top \geq \log \det A_\natural W^\dagger (W^\dagger)^\top A_\natural^\top = \log \det A_\natural A_\natural^\top + \log \det Q^\top W^\dagger (W^\dagger)^\top Q \tag{9}$$

where the first inequality is shown in Appendix A. Regarding the second term, we have

$$\log \det Q^\top W^\dagger (W^\dagger)^\top Q \geq -\log \det Q^\top W^\top W Q \tag{10a}$$

$$\geq -m \log \frac{1}{m}\|Q^\top W^\top\|_F^2, \tag{10b}$$

where (10a) and (10b) are shown Appendix A. Combining (8), (9), and (10) shows that

$$\log \det A_\star A_\star^\top \geq \log \det A_\natural A_\natural^\top. \tag{11}$$

On the other hand, since $(A_\star, S_\star)$ is optimal for (2), we have

$$\frac{1}{2}\log \det A_\star A_\star^\top + \max_{\|d\|_2^2=m} \sum_{c=1}^k d_c\|e_c^\top S_\star\|_1 \leq \frac{1}{2}\log \det A_\natural A_\natural^\top + \max_{\|d\|_2^2=m} \sum_{c=1}^k d_c\|e_c^\top S_\natural\|_1.$$

Lemma 1 and Assumption 1 shows that

$$\max_{\|d\|_2^2=m} \sum_{c=1}^k d_c\|e_c^\top S_\star\|_1 = \max_{\|d\|_2^2=m} \sum_{c=1}^k d_c\|e_c^\top S_\natural\|_1 = m,$$

therefore

$$\log \det A_\star A_\star^\top \leq \log \det A_\natural A_\natural^\top \tag{12}$$

Combining (11) and (12) shows that $A_\natural$ when scaled according to Assumption 1, or any of its column permutation and/or sign flips, is optimal for (2). In Appendix A, we complete the proof that the second requirement of $m$-strongly scattered in the $k$-hypercube guarantees that every solution must satisfy $A_\natural = A_\star D\Pi$, where $D$ is a diagonal matrix with only $\pm 1$ on the diagonal, hence the overcomplete dictionary is identifiable. $\qquad\square$

Our analysis so far has not explicitly mentioned the sparsity of $S_\natural$, which may seem somewhat counter-intuitive. The explanation is two-fold: in the next subsection, we will show that if $S_\natural$ follows a sparse generative model, then cell$(\widetilde{S}_\natural)$ will be $m$-strongly scattered in the $k$-hypercube with very high probability, thus sparsity is implicitly implied in Assumption 3. In particular, our analysis shows that it is necessary that every column of $S_\natural$ contains no more than $m$ nonzeros. On the other hand, Theorem 1 only shows that the dictionary $A_\natural$ is identifiable, but for an overcomplete dictionary it does not necessarily mean that the sparse coefficient $S_\natural$ is identifiable. Fortunately, with the knowledge of the dictionary, the identifiability of the sparse coefficients has been studied extensively (Donoho, 2006; Candès & Wakin, 2008). Here we provide a general result.

**Assumption 4.** Every column of $S_\natural$ contains at most $s$ nonzeros. In addition, $A_\natural$ is a dictionary such that for every $s_0$ with no more than $s$ nonzeros, $s_0$ is the unique solution to the following optimization problem

$$\operatorname*{minimize}_s \ \|s\|_1 \qquad \text{subject to} \ \ A_\natural D_\natural^{-1} s = A_\natural D_\natural^{-1} s_0,$$

where $D_\natural$ is a diagonal matrix with $c$th diagonal defined in (6).

**Corollary 1.** *Consider the overcomplete DL model $X = A_\natural S_\natural$, where $A_\natural \in \mathbb{R}^{m \times k}$ is the groundtruth mixing matrix and $S_\natural \in \mathbb{R}^{k \times n}$ is the groundtruth sparse coefficient matrix. Suppose $A_\natural$ and $S_\natural$ satisfies Assumptions 1–4. Then for any solution of (2), denoted as $(A_\star, S_\star)$, there exist a permutation matrix $\Pi$ and a diagonal matrix $D$ such that $A_\natural = A_\star D\Pi$ and $S_\natural = \Pi^\top D^{-1} S_\star$.*

*Proof.* Theorem 1 shows that $A_\natural$ is identifiable if Assumption 1–3 are satisfied. Assumption 4 guarantees that $S_\natural$ is uniquely determined if $A_\natural$ is given. To see this, we fix $A = A_\natural$ in (2). From Lemma 1, we know that the optimal $d$ should be the diagonal of $D_\natural$ as defined in Assumption 4. Then optimizing (2) with respect to $S$ is equivalent to the following problem with a change-of-variable $\widetilde{S} = D_\natural S$

$$\underset{\widetilde{S}}{\text{minimize}} \quad \|\widetilde{S}\|_1 \qquad \text{subject to} \quad X = A_\natural S_\natural = A_\natural D_\natural^{-1} \widetilde{S}.$$

If every column of $S_\natural$ is at most $s$-sparse, then the optimal $\widetilde{S}_\star = D_\natural S_\natural$, therefore $S_\star = S_\natural$. The result still holds if columns of $A_\natural$ are permuted and/or multiplied with $\pm 1$. $\qquad\square$

## 2.3 SAMPLE COMPLEXITY ANALYSIS

Theorem 1 states that an overcomplete dictionary $A_\natural$ is identifiable if Assumptions 1–3 are satisfied. Assumptions 1 and 2 are quite easy to satisfy, as it is very reasonable to assume that rows of wide matrices $A_\natural$ and $S_\natural$ are linearly independent, and given any $A_\natural$ and $S_\natural$ one can always find the scaling to satisfy (5). The most crucial condition is Assumption 3, or the fact that cell($\widetilde{S}_\natural$) is $m$-strongly scattered in the $k$-hypercube. In this section we assume that a sparse coefficient matrix $S$ is generated from the sparse-Gaussian model, which has appeared in (Wu & Yu, 2017; Wang et al., 2020), and show that in this case Assumption 3 is satisfied with high probability. This is a different generative model than prior works that also use a volume criterion for complete DL (Hu & Huang, 2023a; Sun & Huang, 2024), in which a Bernoulli-Gaussian model is considered. This is because such a generative model cannot guarantee that every column of $S$ is at least $s$-sparse, thus Corollary 1 cannot be invoked to identify both $A_\natural$ and $S_\natural$.

**Assumption 5** (Sparse-Gaussian model). The matrix $S \in \mathbb{R}^{k \times n}$ is generated from a sparse-Gaussian model with parameter $s < k$, denoted as $S \sim \mathcal{SG}(s)$, if every column of $S$ is independently and identically distributed from the following process: a subset $\mathcal{I}$ of size $s$ is uniformly drawn from all size-$s$ subsets of $\{1, \ldots, k\}$, let $s \in \mathbb{R}^k$ be such that $s_i = 0$ if $i \in \mathcal{I}$ and $s_i \sim \mathcal{N}(0, 1)$ if $i \notin \mathcal{I}$, where $\mathcal{N}(0, 1)$ stands for a standard normal distribution.

To check whether $\mathcal{B}_m \subseteq$ cell($\widetilde{S}$), where $\widetilde{S}$ is obtained from scaling its rows to have unit $\ell_1$ norms, it is easier to equivalently check its polar version cell($\widetilde{S}$)$^\circ \subseteq \mathcal{B}_m^\circ$, where the polar of set $\mathcal{S}$ is defined as $\mathcal{S}^\circ = \{x \mid x^\top y \leq 1, \forall y \in \mathcal{S}\}$. For cell($\widetilde{S}$), its polar has a relatively simple form

$$\text{cell}(\widetilde{S})^\circ = \left\{ w \mid \|w^\top \widetilde{S}\|_1 \leq 1 \right\}.$$

The polar for $\mathcal{B}_m$ has a more complicated form

$$\mathcal{B}_m^\circ = \left\{ w \mid \|Q^\top \text{Diag}(\|q_1\|_2, \ldots, \|q_k\|_2)^\dagger w\|_2 \leq 1, \ \forall \ Q^\top Q = I \right\}.$$

Therefore, checking whether cell($\widetilde{S}$)$^\circ \subseteq \mathcal{B}_m^\circ$ is equivalent to checking whether the optimal value of the following problem equals 1:

$$\underset{Q,w}{\text{maximize}} \quad \|Q^\top D^\dagger w\|_2^2 \qquad \text{subject to} \quad \|w^\top \widetilde{S}\|_1 \leq 1, Q^\top Q = I, \tag{13}$$

where $D = \text{Diag}(\|q_1\|_2, \ldots, \|q_k\|_2)$.

**Theorem 2.** *Suppose $S \in \mathbb{R}^{k \times n}$ is generated from the sparse-Gaussian model $\mathcal{SG}(s)$, where $s < m$, and $\widetilde{S}$ is obtained by scaling its rows to have unit $\ell_1$ norm. Then*

$$\Pr\left[ \sup_{\substack{\|w^\top \widetilde{S}\|_1 \leq 1 \\ Q^\top Q = I}} \|Q^\top D^\dagger w\| > 1 \right] \leq 4 \exp\left( \frac{k}{2} \log \frac{k^2}{m} - n \frac{s^2 m}{k^3} \right). \tag{14}$$

*The probability goes to zero exponentially fast as*

$$n = O\left( \frac{k^2}{m} \log \frac{k^2}{m} \right).$$

The proof is relegated to Appendix B. Comparing this result to prior work on complete DL (Hu & Huang, 2023a), in which case the sample complexity is $O(k \log k)$, we see that the bounds agree when $m = k$, which is a good sign that the bound is tight. On the other hand, there is one step in the proof that shows that for overcomplete DL, it is necessary that every column of $S$ is at most $s$-sparse, where $s < m$; this is not required for complete DL. This shows the necessity of adopting the sparse-Gaussian model rather than the Bernoulli-Gaussian model, even if identifiability of $S_{\natural}$ is not required. In fact, even the most relaxed condition on sparse recovery would require $s < m/2$, so assuming $S \sim \mathcal{SG}(s)$ with $s < m$ is a very reasonable assumption in practice.

## 3 Algorithm via Alternating Minimization

We will now design an algorithm for the novel formulation (2) for overcomplete DL, whose global correctness in recovering the ground-truth dictionary has been theoretically established above. The main idea is similar to the (inexact) alternating optimization framework that most DL algorithms adopt. First of all, since most practical applications admit approximate factorization, we move the constraint $X = AS$ in (2) as a penalty term as follows

$$\underset{A,S}{\text{minimize}} \quad \frac{\lambda}{2}\|AS - X\|_{\mathrm{F}}^2 + \frac{1}{2}\log \det AA^\top + \max_{\|d\|_2^2 = m} \sum_{c=1}^{k} d_c \|e_c^\top S\|_1, \tag{15}$$

where the hyper-parameter $\lambda$ balances data fidelity and the identifiability criterion. The rest of this section will focus on designing an iterative algorithm for solving (15). We denote $(A_t, S_t)$ as the updates obtained at the $t$th iteration. In an alternating fashion, $A_{t+1}$ is obtained by fixing $S = S_t$, and $S_{t+1}$ is obtained by fixing $A = A_{t+1}$.

### 3.1 Update of A

Fixing $S = S_t$, the update of $A$ amounts to solving a log-determinant regularized least squares, which is nonconvex optimization problem. We propose two types of updates:

- Gradient descent. As the gradient of $(1/2) \log \det AA^\top$ at $A_t$ is $(A_t^\dagger)^\top$, a simple choice of update is to move along the negative gradient direction with step size $\gamma$

$$A_{t+1} \leftarrow A_t - \gamma \left( \lambda(AS_t - X)S_t^\top + (A_t^\dagger)^\top \right)$$

- Majorization minimization. Notice that the log determinant of a positive definite matrix is concave, therefore

$$\log \det AA^\top \leq \log \det A_t A_t^\top + \mathrm{Tr}[(A_t A_t^\top)^{-1}(AA^\top - A_t A_t^\top)].$$

This defines a quadratic majorization function for $(1/2) \log \det AA^\top$. Minimizing this term plus the fitting term with respect to $A$ amounts to solving the following linear equation:

$$\lambda(AS_t - X)S_t^\top + (A_t A_t^\top)^{-1}A = 0.$$

This is Sylvester's equation, which can be solved by taking the eigen-decomposition of $S_t S_t^\top$ and $A_t A_t^\top$.

For simplicity, we resort to the gradient descent update in the rest of this paper.

### 3.2 Update of S

If $d$ is fixed, then the subproblem of $S$ is a $\ell_1$ regularized least squares problem, which has been extensively studied. It is known to have no closed-form solutions, which is not preferable as one step of an iterative algorithm. Thus, we propose to update $S$ with a proximal gradient step. Since it amounts to getting a linear approximation at $S_t$, we also set $d$ as the optimal choice with $S_t$, which is easy to obtain from Cauchy-Schwarz:

$$d_c = \sqrt{\frac{m}{\sum_{j=1}^{k} \|e_j^\top S_t\|_1^2}} \|e_c^\top S_t\|_1. \tag{16}$$

As a result, the proximal gradient update of $S$ with step size $\gamma$ take the form

$$S_{t+1} \leftarrow \mathcal{T}_{\gamma d} \left( S_t - \gamma \lambda A_{t+1}^\top (A_{t+1} S_t - X) \right),$$

where $\mathcal{T}_{\gamma d}(\cdot)$ is the soft-thresholding operator on a matrix with $k$ rows, and the threshold for components on the $c$th row is $d_c/\gamma$.

### 3.3 SUMMARY AND EXPERIMENTAL DEMONSTRATION

The proposed algorithm based on alternating minimization is summarized in

---

**Algorithm 1** Solving (14) via alternating minimization

---

1: initialize $A_0$ and $S_0$
2: **for** $t = 0, 1, 2, \ldots$ until convergence **do**
3:     $A_{t+1} \leftarrow A_t - \gamma \left( \lambda (A S_t - X) S_t^\top + (A_t^\dagger)^\top \right)$
4:     **for** $c = 1, \ldots, k$ **do**
5:       $d_c = \sqrt{\frac{m}{\sum_{j=1}^k \|e_j^\top S_t\|_1^2}} \|e_c^\top S_t\|_1$
6:     **end for**
7:     $S_{t+1} \leftarrow \mathcal{T}_{\gamma d} \left( S_t - \gamma \lambda A_{t+1}^\top (A_{t+1} S_t - X) \right)$
8: **end for**

---

As (14) is nonconvex and NP-hard, no known algorithm is able to guarantee convergence to a global optimum. In the following, we provide a brief demonstration of the performance of the proposed algorithm. Admittedly, the proposed formulation with identifiability guarantees opens up a new direction for research on algorithm design for dictionary learning, which is a challenging task in itself as it involves several terms that are nontrivial to handle.

We synthetically generate random problems with $s = 5$, $m = 10$, $k = 20$, and $n = 200$. A groundtruth sparse coefficient matrix $S_\natural \in \mathbb{R}^{k \times n}$ is generated from the sparse-Gaussian model $\mathcal{SG}(s)$, while the groundtruth dictionary $A_\natural \in \mathbb{R}^{m \times k}$ is simply generated from a standard normal distribution. The data matrix is then generated as $X = A_\natural S_\natural$. Algorithm 1 runs on $X$ with $\lambda = 1000$ (since we want the data fidelity term to be almost zero) and $\gamma = 10^{-5}$. At the end, both columns of $A_\natural$ and $A_\star$ are scaled to unit $\ell_2$ norm, and the Hungarian algorithm (Kuhn, 1955) is used to find the best column matching. The resulting estimation error $\|A_\natural - A_\star\|_F^2$ remains approximately $10^{-8}$ over multiple runs. Considering the $A_\natural$ is $10 \times 20$ with unit column norms this is a satisfactory initial result.

## 4 CONCLUSION

In this paper, we provide perhaps the first identifiability analysis of a matrix factorization model when the latent dimension is *higher* than the ambient dimension, namely the overcomplete dictionary learning problem. Classical works on this problem rely on combinatorial mathematics, which in turn requires the sample size to be factorial to the latent dimension. Our work is based on a novel formulation that uses a hybrid of weighted $\ell_1$ norm of the sparse coefficients and the volume of the overcomplete dictionary as the identification criterion, and we show that identifiability of the overcomplete dictionary of size $m \times k$ can be guaranteed if a geometric condition is satisfied, namely the cellular hull of the sparse coefficient matrix is $m$-strongly scattered in the $k$-hypercube. If the sparse coefficient matrix is generated from the sparse-Gaussian model, then such identifiability condition can be satisfied with very high probability if the sample size is $O((k^2/m)\log(k^2/m))$, which is a huge improvement compared to prior work with factorial complexity. We also propose an algorithm for the novel overcomplete dictionary learning formulation. The proposed novel formulation for overcomplete DL with a global identifiability guarantee leaves much room for faster and more efficient algorithm design.

### ACKNOWLEDGMENTS

This work is supported in part by NSF ECCS-2237640 and NIH R01LM014027.

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

## A  PROOF OF THEOREM 1

In the proof sketch of Theorem 1, we showed that

$$\frac{1}{2}\log\det A_\natural A_\natural^\top + \max_{\|d\|_2^2=m}\sum_{c=1}^{k}d_c\|e_c^\top S_\natural\|_1 = \frac{1}{2}\log\det A_\star A_\star^\top + \max_{\|d\|_2^2=m}\sum_{c=1}^{k}d_c\|e_c^\top S_\star\|_1,$$

so $(A_\natural, S_\natural)$ is at least one candidate solution for (2). In this section, we first complete the proof by showing that if the second requirement of Assumption 3 is satisfied, then any optimal solution $(A_\star, S_\star)$ must satisfy that $A_\star = A_\natural \Pi D$, where $\Pi$ is a permutation matrix and $D$ is a diagonal matrix with $\pm 1$ on the diagonal.

In (8) we proved that $\|e_c^\top S_\star\|_1 \geq \|Q^\top w_c\|_2$ by choosing $\|\theta\|_\infty \leq 1$ such that

$$\widetilde{S}\theta = D_\natural^{-1}QQ^\top w_c/\|Q^\top w_c\|_2.$$

Suppose $Q^\top w_c/\|Q^\top w_c\|_2 \neq \pm q_c/\|q_c\|_2$ for all $c = 1, \ldots, k$, then according to the second requirement of Assumption 3, it is in the interior of cell$(\widetilde{S}_\natural)$, which means there exists $\alpha > 1$ and $\|\tilde{\theta}\|_\infty \leq 1$ such that

$$\widetilde{S}\tilde{\theta} = \alpha D_\natural^{-1}QQ^\top w_c/\|Q^\top w_c\|_2.$$

Therefore

$$\|e_c^\top S_\star\|_1 = \|w_c^\top S\|_1 \geq w_c D_\natural \widetilde{S}\tilde{\theta} = \alpha\|Q^\top w_c\|_2 > \|Q^\top w_c\|_2.$$

This means equality of $\|e_c^\top S_\star\|_1 \geq \|Q^\top w_c\|_2$ is only attained when $Q^\top w_c = \pm q_c$ for some $c = 1, \ldots, k$.

Finally, it is proven in (10b) that

$$\log\det Q^\top W^\top W Q \leq m\log\frac{1}{m}\|Q^\top W\|_F^2,$$

and if $\det A_\natural A_\natural^\top = \det A_\star A_\star^\top$ the above inequality must hold as an equality. In the proof of (10b) we showed that equality holds only if all the eigenvalues of $Q^\top W^\top W Q$ are equal, meaning columns of $WQ$ are orthonormal. Since $Q$ itself is orthonormal, it is possible only if $WQ = D\Pi^\top Q$, where $D$ is a diagonal matrix with $\pm 1$ on the diagonals and $\Pi$ is a permutation matrix. This shows that

$$A_\star = A_\natural \Pi D.$$

**Q.E.D.**

The remaining of this section shows some key equalities and inequalities in the proof sketch that are skipped for clarity.

*Proof that* $\|e_c^T S_\star\|_1 = \|w_c^\top S_\natural\|_1$. As we argued in the proof sketch of Theorem 1, the constraint $X = AS$ is equivalent to $AW = A_\natural$ and $S = WS_\natural + B$ where $S_\natural B^\top = 0$. Substituting them into (2) eliminates the constraint $X = AS$:

$$\underset{A,W,B}{\text{minimize}}\quad \frac{1}{2}\log\det AA^\top + \max_{\|d\|_2^2=m}\sum_{c=1}^{k}d_c\|w_c^\top S_\natural + b_c^\top\|_1 \qquad \text{subject to}\quad AW = A_\natural, S_\natural B^\top = 0.$$

Taking the Clarke generalized derivative of the Lagrange function with respect to $b_c$ and setting it equal to zero gives

$$d_c\theta_c + S_\natural^\top \mu_c = 0,$$

where $\theta_c = \text{sign}(S_\natural^\top w_c + b_c)$ and $\mu_c$ is the Lagrange multiplier for the $c$th column of $S_\natural B^\top$. This shows that $\theta_c$ is a linear combination of rows of $S_\natural$, which means it is orthogonal to $b_c$. As a result,

$$\|e_c^\top S_\star\|_1 = \|w_c^\top S_\natural + b_c^\top\|_1 = w_c^\top S_\natural \theta_c + b_c^\top \theta_c = w_c^\top S_\natural \theta_c = \|w_c^\top S_\natural\|_1.$$

$\square$

*Proof of inequality* (9). The matrix $\boldsymbol{W}\boldsymbol{W}^\dagger$ defines a projection matrix, which is symmetric, with the following properties:

$$\boldsymbol{W}\boldsymbol{W}^\dagger \preceq \boldsymbol{I}, \quad \boldsymbol{W}\boldsymbol{W}^\dagger = \boldsymbol{W}\boldsymbol{W}^\dagger\boldsymbol{W}\boldsymbol{W}^\dagger = \boldsymbol{W}\boldsymbol{W}^\dagger(\boldsymbol{W}\boldsymbol{W}^\dagger)^\mathsf{T} = \boldsymbol{W}\boldsymbol{W}^\dagger(\boldsymbol{W}^\dagger)^\mathsf{T}\boldsymbol{W}^\mathsf{T}.$$

As a result,

$$\det \boldsymbol{A}_\star \boldsymbol{A}_\star^\mathsf{T} \geq \det \boldsymbol{A}_\star \boldsymbol{W}\boldsymbol{W}^\dagger \boldsymbol{A}_\star^\mathsf{T} = \det \boldsymbol{A}_\star \boldsymbol{W}\boldsymbol{W}^\dagger(\boldsymbol{W}^\dagger)^\mathsf{T}\boldsymbol{W}^\mathsf{T}\boldsymbol{A}_\star^\mathsf{T} = \det \boldsymbol{A}_\natural \boldsymbol{W}^\dagger(\boldsymbol{W}^\dagger)^\mathsf{T}\boldsymbol{A}_\natural^\mathsf{T},$$

where the last step is because $\boldsymbol{A}_\natural = \boldsymbol{A}_\star \boldsymbol{W}$. □

*Proof of inequality* (10a). Since $\boldsymbol{A}_\natural = \boldsymbol{A}_\star \boldsymbol{W}$, rows of $\boldsymbol{A}_\natural$ are in the row space of $\boldsymbol{W}$, and so are columns of $\boldsymbol{Q}$. Therefore

$$\boldsymbol{W}\boldsymbol{W}^\dagger \boldsymbol{Q} = \boldsymbol{Q}.$$

This means $\boldsymbol{W}\boldsymbol{Q}$ has linearly independent columns, therefore $\boldsymbol{Q}^\mathsf{T}\boldsymbol{W}^\mathsf{T}\boldsymbol{W}\boldsymbol{Q}$ is invertible and

$$(\boldsymbol{Q}^\mathsf{T}\boldsymbol{W}^\mathsf{T}\boldsymbol{W}\boldsymbol{Q})^{-1} = (\boldsymbol{W}\boldsymbol{Q})^\dagger(\boldsymbol{Q}^\mathsf{T}\boldsymbol{W}^\mathsf{T})^\dagger.$$

On the other hand, since $\boldsymbol{Q}^\mathsf{T}\boldsymbol{W}\boldsymbol{W}^\dagger\boldsymbol{Q} = \boldsymbol{I}$, this means

$$\boldsymbol{Q}^\mathsf{T}\boldsymbol{W}^\dagger = (\boldsymbol{W}\boldsymbol{Q})^\dagger + \boldsymbol{C},$$

where rows of $\boldsymbol{C}$ are orthogonal to columns of $\boldsymbol{W}\boldsymbol{Q}$, i.e., $\boldsymbol{C}\boldsymbol{W}\boldsymbol{Q} = 0$. Columns of $\boldsymbol{W}\boldsymbol{Q}$ and rows of $(\boldsymbol{W}\boldsymbol{Q})^\dagger$ span the same subspace, so we also have $\boldsymbol{C}^\mathsf{T}(\boldsymbol{W}\boldsymbol{Q})^\dagger = 0$. As a result,

$$\det \boldsymbol{Q}^\mathsf{T}\boldsymbol{W}^\dagger(\boldsymbol{W}^\dagger)^\mathsf{T}\boldsymbol{Q} = \det \left((\boldsymbol{W}\boldsymbol{Q})^\dagger + \boldsymbol{C}\right)\left((\boldsymbol{W}\boldsymbol{Q})^\dagger + \boldsymbol{C}\right)^\mathsf{T}$$

$$= \det \left((\boldsymbol{W}\boldsymbol{Q})^\dagger(\boldsymbol{Q}^\mathsf{T}\boldsymbol{W}^\mathsf{T})^\dagger + \boldsymbol{C}\boldsymbol{C}^\mathsf{T}\right)$$

$$\geq \det(\boldsymbol{W}\boldsymbol{Q})^\dagger(\boldsymbol{Q}^\mathsf{T}\boldsymbol{W}^\mathsf{T})^\dagger = \det(\boldsymbol{Q}^\mathsf{T}\boldsymbol{W}^\mathsf{T}\boldsymbol{W}\boldsymbol{Q})^{-1}.$$

Taking the log on both sides shows (10a). □

*Proof of inequality* (10b). Denote the eigenvalues of $\boldsymbol{Q}^\mathsf{T}\boldsymbol{W}^\mathsf{T}\boldsymbol{W}\boldsymbol{Q}$ as $\lambda_1, \ldots, \lambda_m$, which are all nonnegative, then

$$\left(\det \boldsymbol{Q}^\mathsf{T}\boldsymbol{W}^\mathsf{T}\boldsymbol{W}\boldsymbol{Q}\right)^{1/m} = \left(\prod_{j=1}^m \lambda_j\right)^{1/m} \leq \frac{1}{m}\sum_{j=1}^m \lambda_j = \operatorname{Tr} \boldsymbol{Q}^\mathsf{T}\boldsymbol{W}^\mathsf{T}\boldsymbol{W}\boldsymbol{Q} = \|\boldsymbol{Q}^\mathsf{T}\boldsymbol{W}^\mathsf{T}\|_\mathrm{F}^2,$$

where the inequality in the middle is the geometric-arithmetic mean inequality. Taking the log on both sides and rearranging gives (10b). Notice that equality holds only if $\lambda_1 = \cdots = \lambda_m$, i.e., columns of $\boldsymbol{W}\boldsymbol{Q}$ are orthonormal. □

## B  PROOF OF THEOREM 2

We assume that $\boldsymbol{S} \sim \mathcal{SG}(s)$, and the first thing we do is rescale its rows to have unit $\ell_1$ norms and use it for the problem 13. To simplify the analysis, we can instead directly maximize $\|\boldsymbol{Q}^\mathsf{T}\boldsymbol{D}^{-1}\boldsymbol{w}\|^2$ subject to $\|\boldsymbol{w}^\mathsf{T}\boldsymbol{S}\|_1 \leq 1$, and compare it with the largest $\ell_1$ norm of the rows of $\boldsymbol{S}$. The complement of the intended probability can be bounded as

$$\Pr\left[\sup_{\substack{\|\boldsymbol{w}^\mathsf{T}\tilde{\boldsymbol{S}}\|_1 \leq 1 \\ \boldsymbol{Q}^\mathsf{T}\boldsymbol{Q}=\boldsymbol{I}}} \|\boldsymbol{Q}^\mathsf{T}\boldsymbol{D}^{-1}\boldsymbol{w}\| \leq 1\right] \geq \Pr\left[\sup_{\substack{\|\boldsymbol{w}^\mathsf{T}\boldsymbol{S}\|_1 \leq 1 \\ \boldsymbol{Q}^\mathsf{T}\boldsymbol{Q}=\boldsymbol{I}}} \|\boldsymbol{Q}^\mathsf{T}\boldsymbol{D}^{-1}\boldsymbol{w}\| \leq \alpha \cap \max_j \|\boldsymbol{S}_{j,:}\|_1 \geq \alpha\right],$$

with an arbitrary choice of $\alpha$. Conversely,

$$\Pr\left[\sup_{\substack{\|\boldsymbol{w}^\mathsf{T}\tilde{\boldsymbol{S}}\|_1 \leq 1 \\ \boldsymbol{Q}^\mathsf{T}\boldsymbol{Q}=\boldsymbol{I}}} \|\boldsymbol{Q}^\mathsf{T}\boldsymbol{D}^{-1}\boldsymbol{w}\| > 1\right] \leq \Pr\left[\sup_{\substack{\|\boldsymbol{w}^\mathsf{T}\boldsymbol{S}\|_1 \leq 1 \\ \boldsymbol{Q}^\mathsf{T}\boldsymbol{Q}=\boldsymbol{I}}} \|\boldsymbol{w}\| > \alpha \cup \max_j \|\boldsymbol{S}_{j,:}\|_1 < \alpha\right]$$

$$\leq \Pr\left[\sup_{\substack{\|\boldsymbol{w}^\mathsf{T}\boldsymbol{S}\|_1 \leq 1 \\ \boldsymbol{Q}^\mathsf{T}\boldsymbol{Q}=\boldsymbol{I}}} \|\boldsymbol{Q}^\mathsf{T}\boldsymbol{D}^{-1}\boldsymbol{w}\| > \alpha\right] + \Pr\left[\max_j \|\boldsymbol{S}_{j,:}\|_1 < \alpha\right] \quad (17)$$

where the second inequality is obtained from the union bound. The rest of this section is dedicated to bounding the above two terms. Both of these results rely on the following version of the Bernstein inequality (Bennett, 1962):

**Theorem 3** (Bernstein's inequality). *Let $Z_1, \dots, Z_n$ be independent random variables with $\mathrm{E}[Z_i^2] \le v^2$ and there exists some constant $c$ such that for all integer $d > 2$*

$$\mathrm{E}[|Z_i|^d] \le \frac{1}{2} d! v^2 c^{d-2}. \tag{18}$$

*Then*

$$\Pr\left[\left|\sum_{i=1}^{n}(Z_i - \mathrm{E}[Z_i])\right| > \epsilon\right] \le 2\exp\left(-\frac{\epsilon^2}{2(nv^2 + c\epsilon)}\right)$$

**Lemma 2** (Bounding the second term in (17)). *Suppose $\boldsymbol{S} \in \mathbb{R}^{k \times n}$ is generated from the sparse-Gaussian model $\mathcal{SG}(s)$. Then*

$$\Pr\left[\max_j \|\boldsymbol{S}_{j,:}\|_1 < n(s/k)(\sqrt{2/\pi} - \epsilon)\right] \le 2k\exp\left(-\frac{n(s/k)\epsilon^2}{2 + \sqrt{2}\epsilon}\right)$$

*Proof.* Let $\boldsymbol{s} = (s_1, \dots, s_n)$ be one row of $\boldsymbol{S}$ generated from $\mathcal{SG}(s)$, then each $s_i$ has probability $s/k$ to be standard normal and probability $1 - s/k$ to be zero. We will use Bernstein's inequality with $Z_i = |s_i|$. Let $g$ denote a standard normal random variable, then $|g|$ follows a Chi-distribution of degree 1, so its moments are

$$\mathrm{E}[|g|^d] = 2^{d/2}\frac{\Gamma\left(\frac{d+1}{2}\right)}{\Gamma\left(\frac{1}{2}\right)}. \tag{19}$$

As a result, we have $\mathrm{E}[Z_i] = (s/k)\sqrt{2/\pi}$ and $\mathrm{E}[Z_i^2] = s/k$. Using the recurrence relation for the Gamma function $\Gamma(t+1) = t\Gamma(t)$ and $\sqrt{2}/\Gamma(1/2) = \sqrt{2/\pi} < 1$ we can bound the rest of the moments with $d > 2$ as

$$\mathrm{E}[|Z_i|^d] \le (s/k)\frac{d!}{2^{d/2}}. \tag{20}$$

Therefore the moments satisfy (18) with $c = 1/\sqrt{2}$, results in

$$\Pr\left[\|\boldsymbol{s}\|_1 < n(s/k)(\sqrt{2/\pi} - \epsilon)\right] \le \Pr\left[\left|\sum_{i=1}^{n}(Z_i - \mathrm{E}[Z_i])\right| > n(s/k)\epsilon\right]$$

$$\le 2\exp\left(-\frac{n^2(s/k)^2\epsilon^2}{2(n(s/k) + n(s/k)\epsilon/\sqrt{2})}\right)$$

$$= 2\exp\left(-\frac{n(s/k)\epsilon^2}{2 + \sqrt{2}\epsilon}\right).$$

Finally, using the union bound

$$\Pr\left[\max_j \|\boldsymbol{S}_{j,:}\|_1 < n(s/k)(\sqrt{2/\pi} - \epsilon)\right] \le k\Pr\left[\|\boldsymbol{S}_{j,:}\|_1 < n(s/k)(\sqrt{2/\pi} - \epsilon)\right]$$

$$\le 2k\exp\left(-\frac{n(s/k)\epsilon^2}{2 + \sqrt{2}\epsilon}\right).$$

$\square$

We now proceed to bound the first term in (17). First, we note the following equivalence:

$$\Pr\left[\sup_{\substack{\|\boldsymbol{w}^\top\boldsymbol{S}\|_1 \le 1 \\ \boldsymbol{Q}^\top\boldsymbol{Q}=\boldsymbol{I}}} \|\boldsymbol{Q}^\top\boldsymbol{D}^{-1}\boldsymbol{w}\| > \alpha\right] = \Pr\left[\inf_{\substack{\|\boldsymbol{Q}^\top\boldsymbol{D}^{-1}\boldsymbol{w}\|=1 \\ \boldsymbol{Q}^\top\boldsymbol{Q}=\boldsymbol{I}}} \|\boldsymbol{S}^\top\boldsymbol{w}\|_1 < 1/\alpha\right] \tag{21}$$

We are also going to use the following notion of $\delta$-cover from convex geometry (Pisier, 1999):

**Definition 3** ($\delta$-cover). *A finite $\delta$-cover of a set $\mathcal{S}$ in $\mathbb{R}^k$ is a finite set $\mathcal{N}(\mathcal{S}, \delta)$ of points on $\mathcal{S}$ such that any point on $\mathcal{S}$ is within $\epsilon$ away from an element in $\mathcal{N}(\mathcal{S}, \delta)$, i.e.*

$$\min_{\boldsymbol{w}_i \in \mathcal{N}(\mathcal{S}, \delta)} \|\boldsymbol{w} - \boldsymbol{w}_i\| < \delta, \ \ \forall \boldsymbol{b} \in \mathcal{S}.$$

A well-known result is that the $\delta$-cover of the unit sphere $\mathcal{B}_k = \{\boldsymbol{w} \mid \|\boldsymbol{w}\|_2 = 1\}$ has bounded cardinality (Vershynin, 2018):

$$|\mathcal{N}(\mathcal{B}_k, \delta)| \leq \left(\frac{2}{\delta} + 1\right)^k.$$

Notice that $\mathcal{B}_k$ is a special case of $\mathcal{B}_m$ defined in Assumption 3 with $k = m$, implying that $\mathcal{B}_k \subseteq \mathcal{B}_m$ and thus $\mathcal{B}_m^\circ \subseteq \mathcal{B}_k^\circ$. We will use this result to prove a (rather loose) bound for the $\delta$-cover of $\mathcal{B}_m^\circ$.

**Lemma 3.** *The $\delta$-cover of $\mathcal{B}_m^\circ$ satisfies*

$$|\mathcal{N}(\mathcal{B}_m, \delta)|_2 \leq \left(\frac{2}{\delta} + 1\right)^k$$

*Proof.* It is easy to see that $\mathcal{B}_k$ is self-polar, i.e., $\mathcal{B}_k^\circ = \mathcal{B}_k$, so $\mathcal{B}_m^\circ \subseteq \mathcal{B}_k$. Since the superset $\mathcal{B}_k$ satisfies $|\mathcal{N}(\mathcal{B}_k, \delta)| \leq (1 + 2/\delta)^k$, so is the subset $\mathcal{B}_m^\circ$. $\qquad\square$

**Lemma 4.** *Let $\mathcal{N}(\mathcal{B}_m, \delta) = \{\boldsymbol{w}_i\}$ be a $\delta$-cover for $\mathcal{B}_m^\circ$ in $\mathbb{R}^k$. Assume that we have both the lowerbound*

$$\|\boldsymbol{S}^\top \boldsymbol{w}_i\|_1 \geq \beta, \forall \boldsymbol{w}_i \in \mathcal{N}(\mathcal{B}_m, \delta)$$

*and the upperbound*

$$\|\boldsymbol{S}^\top\|_1 = \sup_{\|\boldsymbol{w}\|_1 \leq 1} \|\boldsymbol{S}^\top \boldsymbol{w}\|_1 \leq \gamma.$$

*Then*

$$\inf_{\boldsymbol{w} \in \mathcal{B}_m^\circ} \|\boldsymbol{S}^\top \boldsymbol{w}\|_1 \geq \beta - \gamma \delta \sqrt{k}$$

*Proof.* By definition of the $\delta$-cover, for all $\boldsymbol{w} \in \mathcal{B}_m^\circ$ we can find $\boldsymbol{w}_i \in \mathcal{N}(\mathcal{B}_m^\circ, \delta)$ with $\|\boldsymbol{w} - \boldsymbol{w}_i\| < \delta$. Therefore

$$\|\boldsymbol{S}^\top \boldsymbol{w}\|_1 \geq \|\boldsymbol{S}^\top \boldsymbol{w}_i\|_1 - \|\boldsymbol{S}^\top(\boldsymbol{w} - \boldsymbol{w}_i)\|_1 \geq \beta - \|\boldsymbol{S}^\top\|_1 \|\boldsymbol{w} - \boldsymbol{w}_i\|_1$$
$$\geq \beta - \|\boldsymbol{S}^\top\|_1 \|\boldsymbol{w} - \boldsymbol{w}_i\|_2 \sqrt{k} \geq \beta - \gamma \delta \sqrt{k}.$$

$\qquad\square$

**Lemma 5** (Bounding the first term in (17)). *Suppose $\boldsymbol{S} \in \mathbb{R}^{k \times n}$ is generated from the sparse-Gaussian model $\mathcal{SG}(p)$. Then*

$$\Pr\left[\inf_{\boldsymbol{w} \in \mathcal{B}_m^\circ} \|\boldsymbol{S}^\top \boldsymbol{w}\|_1 < n(s/k)(\sqrt{2/\pi} - \epsilon) - \delta\sqrt{k} n(s/k)(\sqrt{2/\pi} + \epsilon)\right] \leq \left(\left(\frac{2}{\delta}\right)^k + 2\right) 2 \exp\left(-\frac{n(s/k)^2 \epsilon^2}{2 + \sqrt{2}\epsilon}\right),$$

*where $\delta \in (0, 1)$ represents any choice of $\delta$-cover for $\mathcal{B}_m^\circ$.*

*Proof.* Following Lemma 4, we have

$$\Pr\left[\inf_{\boldsymbol{w} \in \mathcal{B}_m^\circ} \|\boldsymbol{S}^\top \boldsymbol{w}\|_1 < \beta - \gamma \delta \sqrt{k}\right] \leq \sum_{\boldsymbol{w}_i \in \mathcal{N}(\mathcal{B}_m^\circ, \delta)} \Pr\left[\|\boldsymbol{S}^\top \boldsymbol{w}_i\|_1 < \beta\right] + \Pr\left[\|\boldsymbol{S}^\top\|_1 > \gamma\right], \qquad (22)$$

where $|\mathcal{N}(\mathcal{B}_m^\circ, \delta)| < (1 + 2/\delta)^k$ according to Lemma 3.

The bound to the first term in (22) is almost identical to Lemma 2. Dropping the subscript of $\boldsymbol{w}_i$, we write

$$\|\boldsymbol{S}^\top \boldsymbol{w}\|_1 = \sum_{i=1}^n \left|\sum_{j=1}^k s_{ij} w_j\right| := \sum_{i=1}^n |Z_i|.$$

Without the absolute value, $Z_i$ is normally distributed with zero mean and variance $\sigma^2 = \sum_{j \in \mathcal{I}_i} w_j^2$, where $\mathcal{I}_i$ is the index set of nonzero elements in $\boldsymbol{s}_i$ with $|\mathcal{I}_i| = s < m$. Then $\sigma^2 \leq 1$. To see this, suppose without loss of generality that $\mathcal{I}_i = \{1, \ldots, s\}$ and let $\boldsymbol{Q} = [\boldsymbol{I}\ 0]^\top$, then $\boldsymbol{w} \in \mathcal{B}_m^\circ$ implies that $\sum_{j \in \mathcal{I}_i} w_j^2 = \|\boldsymbol{Q}\boldsymbol{w}\|_2 \leq 1$. In other words, if $\boldsymbol{w} \in \mathcal{B}_m^\circ$, the squared sum of no more than $m$ elements of $\boldsymbol{w}$ must be $\leq 1$. Therefore for $d \geq 2$

$$\mathrm{E}[|Z_i|^d] \leq \mathrm{E}[|Z_i|^2] \leq 1,$$

while for $d = 1$ we have

$$\mathrm{E}[|Z_i|] = \frac{s}{k}\sqrt{\frac{2}{\pi}}.$$

Again, the moments satisfy (18) with $c = 1/\sqrt{2}$, results in

$$\Pr\left[\|\boldsymbol{S}^\top \boldsymbol{w}\|_1 < n(s/k)(\sqrt{2/\pi} - \epsilon)\right] \leq \Pr\left[\left|\sum_{i=1}^n (Z_i - \mathrm{E}[Z_i])\right| > n(s/k)\epsilon\right]$$

$$\leq 2\exp\left(-\frac{n^2(s/k)^2\epsilon^2}{2(n + n(s/k)\epsilon/\sqrt{2})}\right)$$

$$= 2\exp\left(-\frac{n(s/k)^2\epsilon^2}{2 + \sqrt{2}\epsilon}\right). \tag{23}$$

To bound $\|\boldsymbol{S}^\top\|_1$, we recall that this is the $\ell_1$ induced norm for matrix $\boldsymbol{S}^\top$, which is shown to be the maximum of the $\ell_1$ norms of the columns of $\boldsymbol{S}^\top$. This means we can use similar arguments used in Lemma 2 (but applied to the other direction) to have

$$\Pr\left[\|\boldsymbol{S}^\top\|_1 > np(\sqrt{2/\pi} + \epsilon)\right] = \Pr\left[\max_j \|\boldsymbol{S}_{j,:}\|_1 > n(s/k)(\sqrt{2/\pi} + \epsilon)\right]$$

$$\leq \Pr\left[\|\boldsymbol{S}_{j,:}\|_1 > n(s/k)(\sqrt{2/\pi} + \epsilon)\right]$$

$$\leq \Pr\left[\left|\sum_{i=1}^n (Z_i - \mathrm{E}[Z_i])\right| > n(s/k)\epsilon\right]$$

$$\leq 2\exp\left(-\frac{n(s/k)\epsilon^2}{2 + \sqrt{2}\epsilon}\right) \leq 2\exp\left(-\frac{n(s/k)^2\epsilon^2}{2 + \sqrt{2}\epsilon}\right), \tag{24}$$

where we pick an arbitrary $j \in [k]$ in the second line since this event implies that the maximum $\ell_1$ norm of the rows is lowerbounded, and in the third line each $Z_i$ satisfies (20). The proof is complete by combining (22), (23), and (24) with $\beta = n(s/k)(\sqrt{2/\pi} - \epsilon)$ and $\gamma = n(s/k)(\sqrt{2/\pi} + \epsilon)$. □

*Proof of Theorem 2.* We first instantiate Lemma 5 with

$$\delta = \frac{n^2(s/k)^2(\sqrt{2/\pi} - \epsilon)^2 - 1}{n^2(s/k)^2(\sqrt{2/\pi} - \epsilon)(\sqrt{2/\pi} + \epsilon)\sqrt{k}},$$

which satisfies $\delta > 0$ if

$$\epsilon < \sqrt{\frac{2}{\pi}} - \frac{1}{n(s/k)}.$$

Then we have

$$\Pr\left[\inf_{\boldsymbol{w} \in \mathcal{B}_m^\circ} \|\boldsymbol{S}^\top \boldsymbol{w}\|_1 < 1/n(s/k)^2(\sqrt{2/\pi} - \epsilon)\right] \leq \left(\left(\frac{2}{\delta}\right)^k + 2\right)2\exp\left(-\frac{n(s/k)\epsilon^2}{2 + \sqrt{2}\epsilon}\right),$$

Combining (17), (21), and Lemma 2 with $\alpha = n(s/k)(\sqrt{2/\pi} - \epsilon)$, we obtain

$$\Pr\left[\sup_{\|\boldsymbol{w}^\top \tilde{\boldsymbol{S}}\|_1 \leq 1} \|\boldsymbol{w}\| > 1\right] \leq \Pr\left[\inf_{\|\boldsymbol{w}\|=1} \|\boldsymbol{S}^\top \boldsymbol{w}\|_1 < 1/\alpha\right] + \Pr\left[\max_j \|\boldsymbol{S}_{j,:}\|_1 < \alpha\right]$$

$$\leq 2\left(k + \left(\frac{2}{\delta}\right)^k + 2\right)\exp\left(-\frac{n(s/k)^2\epsilon^2}{2 + \sqrt{2}\epsilon}\right). \tag{25}$$

Further, with

$$\epsilon < \frac{((m/k)^{1/4} - 1)\sqrt{2/\pi}}{(m/k)^{1/4} + 1},$$

where the right hand side is obviously positive, we have

$$\delta = \frac{n^2(s/k)^2(\sqrt{2/\pi} - \epsilon)^2 - 1}{n^2(s/k)^2(\sqrt{2/\pi} - \epsilon)(\sqrt{2/\pi} + \epsilon)\sqrt{k}} > \frac{n^2(s/k)^2(\sqrt{2/\pi} - \epsilon)^2}{n^2(s/k)^2(\sqrt{2/\pi} + \epsilon)^2\sqrt{k}} > \frac{\sqrt{m}}{k}.$$

We can further relax (25) to

$$\Pr\left[\sup_{\boldsymbol{w} \in \mathcal{B}_m^\circ} \|\boldsymbol{w}\| > 1\right] \leq 2\left(k + \left(2k/\sqrt{m}\right)^k + 2\right)\exp\left(-\frac{n(s/k)^2\epsilon^2}{2 + \sqrt{2}\epsilon}\right)$$

$$\leq 4\left(2k/\sqrt{m}\right)^k \exp\left(-\frac{n(s/k)^2\epsilon^2}{2 + \sqrt{2}\epsilon}\right)$$

$$\leq 4\exp\left((k/2)\log(k^2/m) - n(s/k)^2(m/k)\right),$$

where in the last inequality we simply picked a small enough $\epsilon$ so that

$$\frac{\epsilon^2}{2 + \sqrt{2}\epsilon} < \frac{m}{k}.$$

This completes the proof. □

