# OpenReview forum: "Global Identifiability of Overcomplete Dictionary Learning via L1 and Volume Minimization"
_ICLR.cc/2025/Conference — ICLR 2025 Poster_

### Official Review · Reviewer_LGHw · 2024-10-29

**Soundness:** 2
**Presentation:** 2
**Contribution:** 3
**Rating:** 5
**Confidence:** 3

**Summary:**

This paper presents a novel  formulation for dictionary learning with the dictionary matrix being overcomplete.  Under certain conditions, the authors demonstrate that the novel formulation guarantees global identifiability on the overcomplete dictionary. Finally, the authors design  an alternating optimization algorithm to solve the proposed formulation.

**Strengths:**

It is impressive that the proposed formulation can guarantee  global identifiability over dictionary learning with an overcomplete dictionary matrix under some conditions.

**Weaknesses:**

1. It is not easy to verify whether $A$ and $S$ satisfy the Assumptions 3-4. Hence, it is difficult to evaluate the practical applicability of the theoretical results.
2. The paper provides only a simple simulation experiment, and the results are somewhat unconvincing.
2. The theoretical results are related to the optimal solution to equation 2. However, the proposed optimization algorithm for solving equation 2 cannot  guarantee convergence to a global optimum.

**Questions:**

1. In Lemma 1, it seems that $\Phi=I$ only when the optimal solution to equation 2 is unique. Hence, if there are multiple optimal solutions, does Lemma 1 still hold? If not, how to demonstrate that the optimal solution to equation 2 is unique?
2. How to prove that $A$ in Assumption 4 must exist? In addition, note that $A$ needs to satisfy Assumption 1 as well.
3.  In line 363, the authors state that they aim to check whether the optimal value of equation 12 equals to 1. However, Theorem 2 only gives the probability that the maximum value is greater than 1. What's the relationship between them?
4. Are optimization problems 14 and 2 equivalent? How to determine $\lambda$?
5. For the synthetic experiment, using the estimation error to evaluate the algorithm's performance is somewhat unconvincing. It is more reasonable to show that there exist a permutation matrix and a diagonal matrix that can convert the learned dictionary into the real one. In addition, multiple experiments should be conducted to record the corresponding success probability.
6. Why didn't the authors compare the proposed algorithm with other dictionary learning algorithms in the experiment? Currently, only a simple experiment is available.
7. Where is the Figure mentioned in line 466?
8. Many sentences in Introduction overlap with Hu and Huang (2023a).

---

> ### Author Response · Authors · 2024-11-23
>
> Thank you for your time to assess our work. There seem to be some misunderstanding about the assumptions, so let us clarify them here.
>
> **Assumption 4**
>
> Assumption 4 is exactly the compressive sensing problem, i.e., trying to recover a sparse $s$ from underdetermined linear measurements $x=As$, where $A$ is a wide matrix (and known), via minimizing $||s||_1$. There have been numerous papers about it during late 2000s into early 2010s, most notably pioneered by Donoho and Candes, among many other well-known statisticians. A classical result states that a sparse $s$ is the unique solution if $A$ satisfies the restricted isometry property (RIP); furthermore, if $A$ is randomly generated from standard normal, then it satisfies RIP with high probability if $m=O(s\log s)$. Since it is a well-studied problem, we did not emphasize a lot on this issue, but refer the readers to some seminal papers by Donoho and Candes.
>
> **Assumption 3**
>
> Assumption 3 is original to our paper, and we in fact consider it the biggest contribution. We recognize that it is not easy to verify this condition exactly, which is why we spend the entire subsection $\S2.3$ to justify that if $S$ is generated from a probabilistic model called sparse-Gaussian, Assumption 3 will be satisfied with overwhelming probability if $k=O((m^2/k)\log(m^2/k))$, thus giving some assurance that this is a reasonable assumption as long as the sample size is reasonably large.
>
> **Assumption 1**
>
> In terms of the matrix factorization, Assumption 1 is without loss of generality, because if $X=AS$, we can always put a diagonal matrix $D$ and its inverse $D^{-1}$ in-between to have $X=ADD^{-1}S$. Assumption 1 asks for the diagonal matrix $D$ so that the rescaled $A$ and $S$ satisfies equation (5). The reason we impose this scaling is because, as we showed in Lemma 1, any optimal solution to (2) also satisfies (3), making it easier to analyze identifiability.
>
> **Lemma 1**
>
> Lemma 1 does not require the optimal solution to (2) to be unique. Lemma 1 tries to find out if there are some special properties of any optimal solution in terms of column/row scaling. The argument is that if $(A_\star,S_\star)$ is optimal to (2), then by applying column/row scaling to them is not going to further reduce the objective value of (2). In other words, by plug in $(A_\star\Psi, \Psi^{-1}S_\star)$ into (2) and treating $\Psi$ as the only variable (while $A_\star$ and $S_\star$ are fixed), we cannot further reduce the objective value of (2). Obviously $\Psi=I$ keeps the same objective value, so equivalently it means $I$ minimizes (2).
>
> **Formulation (14)**
>
> While formulation (2) assumes an exact, noiseless model $X=AS$, which is necessary for identifiability analysis, we recognize that in practice it is usually noisy, thus we modify the formulation by moving the constraint $X=AS$ as a data fidelity term $||X-AS||^2$ in (14). This is common practice, for example in compressive sensing people study identifiability by looking at $\min ||s||_1$ subject to $x=As$ but in practice solve the lasso problem $||x-As||^2 + \lambda||s||_1$. The choice of $\lambda$ in principle should reflect the noise level, but in practice has to be tuned.
>
> Since this paper focuses on identifiability analysis, we admit the algorithm design part is somewhat premature. We hope the reviewer could understand that we cannot solve all problems in one paper. We believe the proposed new formulation (2) with identifiability guarantees would inspire many follow-up works, particularly in designing more effective algorithms to solve (2). Thank you again for your invaluable time.

---

> > ### Comment · Reviewer_LGHw · 2024-11-24
> >
> > Many thanks for the authors' response. However, only some of the concerns have been addressed. In addition,
> >
> > 1. If there are multiple optimal solutions to equation (2), there may exist a $\Phi\neq I$ such that $(A_*\Phi,\Phi^{-1}S_*)$ is also an optimal solution. Therefore, I think the proof of Lemma 1 is not rigorous.

---

> > > ### Author Response · Authors · 2024-11-24
> > >
> > > Thank you for your reply. What Lemma 1 tries to show is that as long as $(A_\star,S_\star)$ is *a* solution (possibly one of many), then (3) must hold. It is true that there may exist $\Psi\neq I$ such that $(A_\star\Psi,\Psi^{-1}S_\star)$ is also optimal, but it is *necessary* that $\Psi=I$ is one of the optima. Lemma 1 is looking for a necessary condition for any optimal solution of (2).

---

> > > > ### Comment · Reviewer_LGHw · 2024-11-25
> > > >
> > > > Indeed, $\Psi=I$ is one of the optima and Equation 4 holds only when $\Psi=I$. Does this imply that the solution to equation 2 is unique?

---

> > > > > ### Author Response · Authors · 2024-11-25
> > > > >
> > > > > No it doesn't.
> > > > >
> > > > > Uniqueness is the main goal of this paper, and is formally proven in $\S2.2$; more specifically, Theorem 1 shows that $A$ can be uniquely recovered under Assumptions 1, 2, & 3, and Corollary 1 shows that both $A$ and $S$ can be uniquely recovered under Assumptions 1-4. At Lemma 1, not one assumption has been brought up yet, so uniqueness cannot be achieved (or worse, assumed) at this moment.

---

> > > > > > ### Comment · Reviewer_LGHw · 2024-12-02
> > > > > >
> > > > > > Thank you for answering my questions. I would keep my score, as the experimental results cannot convincingly validate the main claim.

---

### Official Review · Reviewer_BRUJ · 2024-10-31

**Soundness:** 3
**Presentation:** 3
**Contribution:** 3
**Rating:** 6
**Confidence:** 4

**Summary:**

The paper proposes an approach for dictionary learning that uses a loss that mixes a modified, weighted version of the ell-1 norm of the mixture matrix coefficients (with different weights for different rows) with the volume of the dictionary matrix. It identifies a condition for successful identification of the mixing matrix called strong scattering. Similar to existing results, the likelihood of strong scattering for random mixing coefficient matrices such as sparse Gaussian, finding a scaling low for the number of vectors used in learning to scale like $\mathcal{O}\left(\frac{k^2}{m} \log \frac{k^2}{m}\right)$, where $k$ is the number of dictionary elements and $m$ is the data dimension. An alternating minimization algorithm for the proposed optimization is included as well.

**Strengths:**

The formulation appears novel and the analytical results are comprehensive.
A sound identifiability condition is presented.

**Weaknesses:**

As with other conditions for sparse learning and recovery, it appears that the required strong scattering condition cannot be efficiently checked.

It is difficult to assess how much stronger the sufficient scattering condition is versus "that of complete dictionary learning".

Some specific arguments are not clear (see questions).

A figure in the experimental section (cf. Line 466) is missing.

**Questions:**

Line 165: is a square power missing outermost in the second term? Why does this line imply $\alpha = 1$?

Line 171: Why is Assumption 1 reasonable? Is this equality always possible? If so, can that be shown as a lemma?

Line 188: if $\mathcal{B}_m \subseteq \mathcal{S}$, then isn't $\mathcal{B}_m \cap \mathcal{S} = \mathcal{B}$?

Line 246: Assumption 4 has not yet been introduced - can you move the definition earlier?

---

> ### Author Response · Authors · 2024-11-23
>
> Thank you for your positive assessment. Regarding some of your concerns:
>
> > As with other conditions for sparse learning and recovery, it appears that the required strong scattering condition cannot be efficiently checked.
>
> Section $\S2.3$ is dedicated to answer this question. Although checking this condition exactly is hard, a randomly generated $S$ from the sparse-Gaussian model will satisfy the strong scattering condition with very high probability as long as the sample size is more than $O((k^2/m)\log(k^2/m))$. We hope the analysis in $\S2.3$ gives readers some assurance that this is indeed a reasonable assumption in practice.
>
> >It is difficult to assess how much stronger the sufficient scattering condition is versus "that of complete dictionary learning".
>
> An intuitive illustration of how much stronger the scattering condition is shown in Figure 1. Another way to see it is to check the sample complexity analysis. For complete dictionary learning with $k\times k$ dictionaries, Hu and Huang [2023] showed that it requires $O(k\log(k))$ samples. In this paper with overcomplete $m\times k$ dictionaries, our sample complexity is $O((k^2/m)\log(k^2/m))$. The sample complexity is indeed larger, but not by a lot.
>
> ## Questions:
> - Line 165: This line gives $\alpha \sum(\cdots) =m$, while in line 161 we have $\alpha^2\sum(\cdots)=m$, so these two equations imply $\alpha=1$.
> - Line 171: Yes, it is always possible to assume Assumption 1 by rescaling the columns of $A_\natural$ and counter-scale the rows of $S_\natural$. The proof will be identical to that of Lemma 1.
> - Line 188: This is indeed a typo. It should have been $\partial\mathcal{S}$, i.e., the boundary of $\mathcal{S}$. We have fixed it in the revision.
> - Line 246: Indeed it should not have appeared here. Thank you for noticing. We have fixed it in the revision.

---

### Official Review · Reviewer_z7Po · 2024-11-03

**Soundness:** 3
**Presentation:** 3
**Contribution:** 3
**Rating:** 6
**Confidence:** 3

**Summary:**

This paper addresses the identification problem in over-complete dictionary learning by introducing a new formulation. The authors primarily build on the analysis from [Huang & Hu, 2023], extending the concept of "sufficiently scattered" to the over-complete setting. By combining this extension with scaling and independence conditions for $A$ and $S$, the authors argue that "sufficiently scattered" serves as a sufficient condition for the identifiability of $A$ under the proposed formulation (2). Additionally, they provide a theoretical guarantee that this "sufficiently scattered" condition holds with high probability under the commonly used Bernoulli-Gaussian distribution.

**Strengths:**

The idea is well-motivated, and the problem is relevant to the community. While previous work typically relies on column incoherence for $A$, the authors propose a novel sufficient condition of $S$ for the global identifiability of the over-complete dictionary learning problem under their formulation. This is achieved by extending the "sufficiently scattered" condition from non-negative matrix factorization (NMF) to the context of dictionary learning.

**Weaknesses:**

1) The connection between the proposed "sufficiently scattered" condition and the conditions outlined in [3] remains unclear. Could the authors clarify this relationship?

2) The paper appears to be incomplete. For instance, the figure for the experimental section is missing, and in line 187, it seems that $\mathcal{S} \subseteq \mathbb{R}^k$ should be used.

**Questions:**

Given that the "sufficiently scattered" condition has been previously introduced in NMF and topic modeling, and that similar identifiability conditions appear in [1,2], could the authors discuss the specific technical challenges posed by applying this condition in the dictionary learning (DL) setting compared to the NMF/topic modeling context?

[1] Kejun Huang, Nicholas D Sidiropoulos, and Ananthram Swami. Non-negative matrix factorizationrevisited: Uniqueness and algorithm for symmetric decomposition. IEEE Transactions on Signal Processing, 62(1):211–224, 2013.

[2] Kejun Huang, Xiao Fu, and Nikolaos D Sidiropoulos. Anchor-free correlated topic modeling: Identifiability and algorithm. Advances in Neural Information Processing Systems, 29, 2016.

[3] P. Georgiev, F. Theis and A. Cichocki, "Sparse component analysis and blind source separation of underdetermined mixtures," in IEEE Transactions on Neural Networks, vol. 16, no. 4, pp. 992-996, July 2005

---

> ### Author Response · Authors · 2024-11-23
>
> Thank you for your positive assessment. Let us clarify the differences between some related works:
>
> - Reference [3] is more similar to [Aharon et al., 2006b], [Hillar & Sommer, 2015], and [Cohen & Gillis, 2019], i.e., sparsity is imposed by directly minimizing the number of nonzeros of $S$. Their results are similar too: the Kruskal's rank (or spark) of the dictionary is big enough (which is a mild assumption), and the sparse coefficient matrix $S$ contains enough combinations of the sparsity patterns. The latter is a very strong assumption, which requires a sample size that is factorial in $k$ and $s$.
>
> - The sufficiently scattered condition in NMF is quite different from this paper. Since NMF assumes the sources to be nonnegative, the sufficiently scattered condition in [1,2] is an assumption that is applied to a set in the *nonnegative orthant*. There is a highly related version that applies to a set in the probability simplex. Dictionary learning is not constrained to be nonnegative, so the results in [1,2] cannot be directly applied. In [Hu & Huang, 2023a], a sufficiently scattered condition is proposed for complete dictionary learning, but this one applies to a set in the *hypercube*, which is illustrated in Figure 1(left). For overcomplete dictionary learning, which is the focus of this paper, the sufficiently scattered condition in the hypercube in [Hu & Huang, 2023a] is not enough to guarantee identifiability. The strongly scattered condition proposed in this paper, as illustrated in Figure 1(right), is shown to guarantee identifiability.
>
> - In line 187, it is correct to write $\mathcal{S}\in\mathcal{C}_k$, because we are only interested in sets that are contained in the hypercube. We will fix the figure in the experimental section too. Thanks for your careful review.

---

### Official Review · Reviewer_1DsL · 2024-11-12

**Soundness:** 3
**Presentation:** 3
**Contribution:** 3
**Rating:** 8
**Confidence:** 3

**Summary:**

The paper introduces a new formulation for the overcomplete dictionary learning problem. The authors show global identifiability of the dictionary and sources up to permutation and scaling provided that the atoms are sufficiently sparse.

**Strengths:**

The paper seems mathematically sound (be careful with the dimensions, see the detailed comments below). Its positioning with respect to the existing literature should be better documented though. Two results appear as particularly related: Hu and Huang 2023 and Agarwal et al/ Rambhatla et al. It would help to have a clear discussion on the improvement of the paper compared to those results.

**Weaknesses:**

Sparse coding or sparse dictionary learning are not new

**Questions:**

Detailed Comments:

- Maybe recall what complete and overcomplete (no orthogonality) dictionary mean
- Formulation (2) should be better introduced. Why is
- line 106, you say that A should be a dictionary that guarantees exact recovery of all s-sparse vectors. Do you mean that min ||x||_1 s.t y= Ax should have a unique solution for all s-sparse vectors?
- line 107, what is the cellular hull?
- you should clarify the notion of scattered cellular hull before introducing your results.
- Statement of Lemma 1 is misleading. First of all, from what I understand the weights d_{*c} reach the maximum of \sum_c d_c ||e_cS_*|| under the constraint \|d\|\leq m. Secondly, if the max is attained for (3), why not just optimize the l1 norm squared?
- Is it always possible to scale the columns of A_{\#} and rows of S_{\#} to satisfy (5). This is not obvious to me
- If I understand well you want the set S to be reduced to canonical vectors p? and S could include vectors that are not in the span of Q but all vectors in span(Q) must be of the form q/||q||?
- From your definition of B_m, the set is a subset of R^k (i.e. it is given by some linear combinations of the columns of Q). Moreover S is also a subset of R^k so how can the intersection of those subsets be a subset of R^m (i.e given by rows of Q)? Maybe you mean the columns of Q?
- line 158-159, I would add just one sentence, to explain that for the correlation to be maximum, you need the cosine of the angle between the vectors to be maximum which implies d_c = \alpha \|e_c^T S_*\| for all c
- lines 164-165, there are alphas missing.
- line 178 and Figure 1. If I understand well, the set B_m is an intersection of spheres of dimension m. If my understanding is correct, I think it would be worth mentioning it somewhere because it looks as if the points clouds in Fig 1 have non empty inerior (especially the 2-strongly scattered one) while my guess is there are empty.
- lines 241-244, in your proof sketch, again if I understand well, you define your matrix Q from the left factor of the SVD of A_#. I.e. if you have A_# = U\Sigma V^T, then you define Q as V. Then why not say it like that. I feel this is simpler and much more clear
- On line 248, you refer to assumption 4 which does not appear anywhere (the hyperlink does not work)
- line 251-253, shouldn’t the pseudo inverse be applied on the right of S_*, i.e. from line 252, the dimensions of W seems to be n\times n to me. Moreover, what you need to project to have the decomposition of line 251 are the rows of S_* not the columns.
- One lines 268-269, if I’m not wrong you mulyiply both sides by S_# and not S_*
- On line 272, there is a transpose missing on the second A_#
- On line 272, the last equality in Equation (8) is not completely clear to me. Isn’t ||e_c^T S_*||_1 = ||w_c^T S_#||_1 and not ||e_c^T S_#||_1 ? why is ||w_c^T S_#||_1 = ||e_c^T S_#||_1 ? Does the relation follow from (5) and the fact that A_# = A_*D\Pi ? It would help to have even a short additional explanation here.
- lines 303-305, I don’t understand the sentence. You say that the sparsity is implicitely implied in (5)? How come ?
- lines 302 - 303 should be rephrased. I think what you mean is that “sparsity is required to have the strongly scattered condition used in the statement of Theorem 1” instead of “sparsity is implied in Assumption 1”
- line 308 “does not necessarily mean that the sparse coefficients S_# is identifiable” —> “are identifiable” ?
- lines 313 -320, Assumption 4 seems quite strong (or quite vague) on the dictionary. Is it easy to find such dictionaries? (I.e. you don’t provide any numerical illustration). It would be perhaps good to have a short comment such as the one at the beginning of section 2.3
- lines 339-340, “the most crucial condition is assumption 3 that cell ..” —> “the most crucial condition is assumption 3, or the fact that cell(S_#) should be generated …”?
- lines 341-342: “and show that when it satisfied assumption 3” —> do you mean “and show that it satisfies assumption 3”?
- Section 2.3., lines 337-346, I don’t really understand why, if you can make it work in the sparse Gaussian model, you can’t make it work in the Bernoulli Gaussian model. If the probability in the Bernoulli distribution is set to s/n, can’t you get a result similar to what you have with sufficient probability? Even if you can’t be at least s-sparse, isn’t “at least s-sparse” with sufficient probability enough?
- line 348 - 349 “if for every column of S” —> “if every column of S”
- lines 362-363 : “is equal to” or “equals” but not “equals to”
- line 380, I would remove the line “which is a good sign that the bound is tight ”
- line 383 “even if identifiability of S_# is not required”, what do you mean “is not required”? Aren’t all your result focusing on the identifiability of S_# ? i would remove the paragraph starting from “On the other hand” because it makes everything unclear.
- line 389 - 390, the sentence “Due to the novel formulation (2) for overcomplete …” does not make sense either. Do you mean “We will now design an algorithm for formulation (2) for which uniqueness (up to permutation and scaling) of the dictionary and sources was shown above”
- line 427 “which is not preferable as one step of an iterative algorithm ” just remove.

---

> ### Author Response · Authors · 2024-11-27
>
> Thank you for your positive assessment and careful reading. We will carefully revise the paper according to your constructive comments. Let us address a few comments that may benefit from some responses:
> - Line 106: The reviewer's understanding is correct. We want the proposed formulation to be closely related to the compressive sensing problem, so that for the factorization model $X=AS$, if $A$ is uniquely recovered, then $S$ can be uniquely recovered as well. This is formally addressed in Corollary 1. This also relates to the discussion about Assumption 4, and yes, from the vast literature on compressive sensing, it is quite easy to obtain such a dictionary (for example, by randomly generating $A$ from i.i.d. Gaussian with $m=O(s\log(k))$).
> - Statement of Lemma 1: Indeed the formulation could also be $\ell_1$ norm squared and Theorem 1 could still hold to identify $A$, but then to recover $S$ from the correct $A$, it is not clear whether optimizing $\ell_1$ norm squared is able to identify $S$. By making it a weighted sum of $\ell_1$ norms, it is possible to transform it into a compressive sensing problem and thus numerous prior results can be applied.
> - It is always possible to satisfy (5), by replacing $(A_\star,S_\star)$ with $(A_\natural,S_\natural)$ in the proof of Lemma 1.
> - Definition of $\mathcal{B}_m$: this was indeed a rather serious typo. It should be $\Psi Q q/||q||$, not simply $q/||q||$. Thank you for your careful reading.
> - Line 272: Suppose the QR factorization of $A_\natural^T =QR$ then $\log\det(A_\natural W^{\dagger}(W^{\dagger})^T A_\natural^T) = \log\det R^T + \log\det(Q^T W^{\dagger}(W^{\dagger})^T Q) + \log\det R = \log\det(A_\natural A_\natural^T) + \log\det(Q^T W^{\dagger}(W^{\dagger})^T Q)$.
> - Line 303-305: that was indeed a typo. We mean sparsity is implicitly implied by Assumption **3**.
> - sparse-Gaussian (SG) vs. Bernoulli-Gaussian (BG): In one step of the full proof, specifically line 865-866, we found that we do need the sparsity of every column of $S$ to be at most $s$, so SG model becomes much more handy to work with than BG. We suppose for BG we would need one more term in the bound to exclude the probability that $S$ contains even one column with more than $s$ nonzeros, and the resulting bound may not be so clean.
> - Line 383: Theorem 1 shows that $A$ is identifiable under Assumption 1-3, and in this case $S$ is not necessarily identifiable (we need an additional Assumption 4 according to Corollary 1). We speculate that in practice, it is possible that one is only interested in finding the correct dictionary, but the specific sparse coefficients for all samples is not that important (maybe they are treated as training data and the user is more interested in checking the performance of the dictionary on some test samples), hence the sentence “even if identifiability of $S_\natural$ is not required” (but requires identifiability of $A_\natural$).

---

### Meta-Review · Area_Chair_22Tt · 2024-12-28

**Metareview:**

The paper studies the identifiability of sparse dictionary models in both the complete and overcomplete case. It derives deterministic conditions for global identification (up to permutation and scale), and shows that under a random-sparse-Gaussian model, an m x k dictionary is identifiable with high probability when the number of observations exceeds k^2 / m. The proposed approach studies the global minimizer of a novel objective function which combines the volume log det (AA’) spanned by the rows of the dictionary matrix and the maximum L1 norm of the rows of the sparse coefficient matrix S, and argues that when the coefficients are “sufficiently scattered” on the hypercube, the target factorization is unique.

As described above, the paper gives a novel sufficient condition for global identifiability of sparse dictionary models (the sufficiently scattered condition). When applied to random coefficient models, this result significantly improves over existing identifiability results for overcomplete dictionary learning. Interestingly, the analysis guarantees the correct recovery of the dictionary even for A which L1 minimization does not recover all s-sparse coefficient vectors. The main limitation of the work is that it only guarantees identifiability - not recovery by a tractable algorithm. With that said, identifiability of sparse models remains a fundamental problem in representation learning; the paper advances the understanding of this problem with a novel condition and better rates for overcomplete models.

**Additional Comments On Reviewer Discussion:**

Reviewers praised the paper’s mathematical soundness, noting that it gives a novel sufficient condition for global identifiability of sparse dictionary models, and noted connections between the sufficiently scattered condition for dictionary learning and corresponding conditions for nonnegative matrix factorization. The discussion clarified this connection, as well as the novelties of the paper wrt the literature on NMF.

Two main limitations were noted by reviewers. First, the sufficiently scattered condition may be challenging to check or verify in practice [BRUj,LGHw]. As the reviewers note, this condition can be verified mathematically for random coefficient models, but cannot be checked experimentally on real data. The second limitation of the work is that the paper does not provide computational guarantees: the theory verifies that the global optimum of the proposed objective function corresponds to the target dictionary. However, the proposed optimization formulation is nonconvex, and is not guaranteed to find a global optimum [BRUj].

Reviewers also noted that while this is mostly a theoretical paper, the experimental section could be improved (and is seemingly missing a figure).

---

### Decision · Program_Chairs · 2025-01-22

Accept (Poster)